# Inhibition Effects of *Nippostrongylus brasiliensis* and Its Derivatives against Atherosclerosis in ApoE^-/-^ Mice through Anti-Inflammatory Response

**DOI:** 10.3390/pathogens11101208

**Published:** 2022-10-20

**Authors:** Yougui Yang, Xin Ding, Fuzhong Chen, Xiaomin Wu, Yuying Chen, Qiang Zhang, Jun Cao, Junhong Wang, Yang Dai

**Affiliations:** 1School of Public Health, Nanjing Medical University, Nanjing 211166, China; 2Key Laboratory of National Health Commission on Parasitic Disease Control and Prevention, Key Laboratory of Jiangsu Province on Parasite and Vector Control Technology, Jiangsu Institute of Parasitic Diseases, Wuxi 214064, China; 3Department of Cardiology, the First Affiliated Hospital of Nanjing Medical University, Nanjing 210029, China; 4Microbiological Laboratory, Anhui Provincial Center for Disease Control and Prevention, Hefei 230601, China

**Keywords:** atherosclerosis, *Nippostrongylus brasiliensis*, hookworm, intervention, protective effect, apolipoprotein-E-deficient mouse

## Abstract

Atherosclerosis (AS) is a dominant and growing cause of death and disability worldwide that involves inflammation from its inception to the emergence of complications. Studies have demonstrated that intervention with helminth infections or derived products could modulate the host immune response and effectively prevent or mitigate the onset and progression of inflammation-related diseases. Therefore, to understand the molecular mechanisms underlying the development of atherosclerosis, we intervened in ApoE^-/-^ mice maintained on a high-fat diet with *Nippostrongylus brasiliensis* (*N. brasiliensis*) infection and immunized with its derived products. We found that *N. brasiliensis* infection and its derived proteins had suitable protective effects both in the initial and progressive stages of atherosclerosis, effectively reducing aortic arch plaque areas and liver lipid contents and downregulating serum LDL levels, which may be associated with the significant upregulation of serum anti-inflammatory cytokines (IL-10 and IL-4) and the down-regulation of proinflammatory cytokines (TNF-α and IFN-γ) in the serum. In conclusion, these data highlighted the effective regulatory role of *N. brasiliensis* and its derived proteins in the development and progression of atherosclerosis. This could provide a promising new avenue for the prevention and treatment of atherosclerosis.

## 1. Introduction

Atherosclerosis (AS) is a type of low-grade chronic inflammatory disease characterized by abnormal lipid metabolism and the infiltration of inflammatory cells in arterial walls; this is a major global cause of myocardial infarction and stroke [1]. Mechanisms of atherosclerosis initiation and progression have yet to be fully elucidated, although various risk factors have been identified, including heredity, hypertension, cigarette smoking, and diabetes mellitus; these factors can all cause atherosclerosis and thrombotic complications [2]. Recent studies showed that the balance between proinflammatory and inflammatory-resolving mechanisms plays a vital role in the atherosclerosis process and can dictate the final clinical outcomes [1,3,4]. Various subsets of monocyte-derived macrophages, T lymphocytes, and mast cells are known to participate in different stages of atherosclerosis, including lipid deposition in the endarterium, plaque growth, foam cell formation, and plaque rupture [5]. Therefore, inhibiting proinflammatory effects has emerged as a promising therapeutic approach to improve current lipid-lowering treatments [3,6,7].

Epidemiological investigations identified a negative relationship between helminth infection and the incidence of inflammatory diseases and were defined as the “hygiene hypothesis” by Strachan as early as 1989 [8]. Based on this hypothesis, numerous studies have used helminth infection (hookworm, whipworm, and schistosomes) or helminth-derived molecules for the intervention of allergic asthma, ulcerative colitis, and other inflammatory diseases [9,10,11]. Results showed that interventions through helminth infection or derived products could modulate host immune responses and further prevent or alleviate the initiation and progression of inflammation-related diseases [11,12,13,14]. Furthermore, recent studies showed that infection by some helminths (hookworm, whipworm, and liver fluke) or derived products could also modulate glucose and lipid metabolism in the host, manifesting as weight loss, a reduction in BMI (body mass index, BMI) lower total cholesterol levels and reduced insulin resistance, thus indicating that helminth infection or derived products can exert immune modulation and anti-inflammation effects [15,16]. These previous studies showed that the interventional effects of helminth infection or its derived products were positive for diseases related to low-grade chronic inflammation and could provide a new approach to the prevention and treatment of such diseases.

Hookworm, a type of soil-transmitted helminth, is mainly endemic in developing countries within tropical and subtropical regions worldwide [17]. Estimates showed that 438.9 million people were infected globally by these organisms, resulting in 4.1 million disability-adjusted life years (DALY) and a loss of over 100 billion U.S. dollars annually [18,19]. Human hookworm, mainly including *Ancylostoma duodenale*, *Necator americanus,* and *Ancylostoma ceylanicum*, can cause chronic blood loss from their digestive tract mucosa and severe anemia in cases of heavy infection [17]. A previous meta-analysis relating to helminth infection and human metabolic syndrome showed that human helminth infections, including hookworm infection, could prevent the development of metabolic diseases by reducing glucose levels and insulin resistance, thus halving the rate of human metabolic dysfunction [20]. Furthermore, a significant increase in insulin resistance was observed among helminth-infected subjects following anti-helminthic treatment [21]. Furthermore, a cross-sectional study showed that helminth infections, including hookworm, were negatively associated with risk factors of atherosclerosis manifested by a low BMI, waist:hip ratio, total cholesterol, and LDL-cholesterol levels [22]. Collectively, these human studies indicated that hookworm infection might be beneficial for metabolic dysfunction, including AS. However, the detailed antiatherosclerosis mechanisms involved have yet to be elucidated. *Nippostrongylus brasiliensis* (*N. brasiliensis*), also known as rat hookworm, can infect rodent animals and exhibits a similar life cycle and infection route as human hookworm, thus making it an ideal model for hookworm research [23]. Previous research showed that *N. brasiliensis* infection could induce type two immune responses, manifesting as a high level of anti-inflammatory cytokines, including IL-4, IL-10, and IL-13, which could play a role in tissue repair and the removal of worms in a rodent model [24]. Our previous study further confirmed that products derived from *N. brasiliensis*, larvae 3 stage (L3)- and larvae 5 stage worm (L5)-derived proteins, could induce an alternative activated macrophage phenotype with anti-inflammation properties in vitro [25]. Based on previous studies, we hypothesized that hookworm (*N. brasiliensis*) or its derived products might generate positive effects on AS through induced host anti-inflammation responses.

To test this hypothesis, two independent animal experiments were designed by using the ApoE^-/-^ mouse model maintained on a high-fat diet; one experiment targeted AS initiation, while the other targeted the processes involved in progression. Following intervention with *N. brasiliensis* infection and immunization with its derived products, we analyzed several factors to evaluate interventional effects, including body weight, histopathological alterations in the liver and aortic arch, along with serum levels of lipids and cytokines.

## 2. Materials and Methods

### 2.1. Parasites and Animals

*N. brasiliensis* was gifted by Professor Alex Loukas at James Cook University and has been bred and maintained in our laboratory for approximately five years. Sprague-Dawley rats (male, 300 g in weight) were purchased from Gempharmatech Co., Ltd. (Nanjing, China) and raised by the Animal Center of Jiangsu Institute of Parasitic Diseases (JIPD, Wuxi, China) and used as the definitive host for parasite breeding in accordance with a previous protocol [23]. Briefly, *N. brasiliensis* was maintained by infecting rats with L3 stage larvae by subcutaneous injection and collecting feces daily from day 6 to day 9. The feces were mixed with appropriate amounts of 5 μg/mL amphotericin solution, carbon powder, and corn cob bedding to form a paste, transferred to absorbent paper in the center of a Petri dish, and incubated at 26 °C. After 1 w, L3 larvae were observed to migrate to the edge of the Petri dish, collected, and washed in PBS.

### 2.2. Preparation of Proteins Derived from N. brasiliensis

Proteins derived from the L3 and L5 stage larvae of *N. brasiliensis* were used for the evaluation of efficacy. L3-stage larvae were collected from the Petri dishes of cultured and infected rat fecal samples, as described previously [23]. L5 stage larvae (pre-adult worms) of *N. brasiliensis* were collected from the small intestine of infected rats as described previously [26]. Proteins derived from L3 and L5 larvae were extracted by grinding under cold conditions in parallel with repeated freezing and thawing. Following centrifugation at 10,000× *g* for 10 min at 4 °C, the supernatant was collected, quantified with a BCA Protein Assay Kit (ThermoFisher, Waltham, MA, USA), and stored at −80 °C to await further analysis.

### 2.3. The Mouse Model of Atherosclerosis and In Vivo Interventions

The apolipoprotein-E-deficient (ApoE^-/-^) mouse was used to establish a mouse model of AS; this was induced by feeding with a high-fat diet (1.25% cholesterol) which as purchased from Research Diets (D12108C, Nanjing, China). For the AS initiation intervention experiment (Figure 1A and Figure 2A), forty male ApoE^-/-^ mice (5 weeks old) were provided by the Animal Center of JIPD; these were divided randomly into five groups with 8 mice in each group: model group (model), *N. brasiliensis* infection group (infection), vehicle control group (vehicle), L3 protein group (L3-protein) and L5 protein group (L5-protein). All mice were fed a high-fat diet for 12 weeks. Five hundred L3 larvae were injected subcutaneously into each mouse from the infection group every two weeks. In addition, extracted L3 and L5 proteins were emulsified with an equal volume of Freund’s incomplete adjuvant (Sigma-Aldrich, Shanghai, China) and injected subcutaneously every two weeks (100 micrograms per mouse, with six times totally) into mice in the L3-protein and L5-protein groups, respectively. The same amount of Freund’s incomplete adjuvant was used for each mouse in the vehicle control group with the same time interval. All mice were euthanized at the end of week 12 to evaluate the effects of the intervention.

For the AS progression intervention experiment (Figure 3A and Figure 4A), another 40 male ApoE^-/-^ mice (5 weeks old) were used and randomly divided into five groups (eight mice per group): model, infection, vehicle, L3-protein, and L5-protein groups. All mice were fed a high-fat diet for 20 weeks. Beginning at 12 weeks, each mouse from the five groups was treated with the same amounts, same injection routes, and intervals (with four times total) as described for the AS initiation experiment. All mice were euthanized at the end of week 20 to evaluate the effects of the intervention.

### 2.4. Histopathological Analysis

Tissues, including the aortic arch and liver, from all mice from the two experiments were harvested carefully after euthanasia and used for histopathological analysis, which covered plaque areas of AS and lipid contents in the plaque and liver. For aortic arch tissues, we cut 5 μm sections that spanned the entire aortic arch (20 sections in total). For liver tissues, the same part of liver tissue was collected from each mouse, and 5 μm slices were cut into 10 sections. Oil Red O staining was carried out for each slide with the Oil Red O Staining Kit (Beyotime, Shanghai, China) in accordance with the manufacturer’s instructions. The stained slides were observed by microscopy with low-power and high-power fields (BX53; Olympus, Beijing, China) and photographed by Cellsens Dimension software (https://www.olympus.com.cn/) (accessed on 12 July 2022). The photographed images were analyzed, and relative AS plaque area and lipid contents were calculated through ImageJ software (https://imagej.nih.gov/ij/) (accessed on 15 April 2022). For the lipid content of the liver, 3–5 microscopic fields per mouse were selected. A percentage of red lipid droplets over the entire field was quantified and used for statistical analysis.

### 2.5. Detection of Serum Cytokine Levels

Following euthanasia at the end of the two experiments, blood samples were obtained from the eyeballs of each mouse, and serum samples were isolated by centrifugation at 3000× *g* for 15 min at 4 °C. Levels of cytokines, including IL-2, IFN-γ, TNF-α, IL-6, IL-4, and IL-10, were detected by flow cytometry (BD FACSVerse, Franklin, TN, USA) with Cytometric Bead Array (CBA) Flex Sets Kit (BD Pharmingen, Shanghai, China) according to the kit protocols.

### 2.6. Detection of Serum Lipid Levels

In the AS progression intervention experiment, blood samples were obtained from the eyeballs of each mouse at the end of week 20; the animals had fasted for one day previously. Serum was isolated, as described above, for the detection of lipid levels, including total cholesterol, triglyceride, and low- and high-density lipoproteins, by commercial kits (Solarbio Life Sciences, Beijing, China) according to the kit instructions.

### 2.7. Statistical Analysis

GraphPad Software (Version 6.01) was applied for data processing and statistical analysis. Data comparison involved the Student’s *t*-test or one-way ANOVA followed by Dunnett’s multiple comparisons. *p* < 0.05 was considered statistically significant.

## 3. Results

### 3.1. The effects of N. brasiliensis Infection and Derived Products on the Initiation of Atherosclerosis

#### 3.1.1. Body Weight Changes

The weight of each mouse in the five groups was measured every 2 weeks. As shown in Figure 1B and Figure 2B, body weight increased gradually when fed a high-fat diet (HFD). Compared to the HFD model group, mice from the *N. brasiliensis* infection group showed a slower body weight gain. A significant reduction in body weight in the *N. brasiliensis* infection group was observed when compared to that in the model group in week 12 (*p* < 0.05, Figure 1B). Meanwhile, compared to the vehicle group, mice from the L3-protein and L5-protein groups also showed a slower body weight increase. When compared to the vehicle group in week 12, a significant decrease in body weight in the L3-protein and L5-protein groups was observed (*p* < 0.05, Figure 2B). This indicated that ApoE^-/-^ mice fed a high-fat diet experienced a loss of body weight following the initiation of AS by *N. brasiliensis* infection and intervention with derived products.

#### 3.1.2. Histopathological Alterations in the Liver and Aortic Arch

Relative lipid content in the liver was detected by the Oil Red O staining of pathological sections. As shown in Figure 1C,D, there was a significant decrease in lipid content in the *N. brasiliensis* infection group when compared to that in the model group (*p* < 0.0001). Furthermore, a significant decrease in lipid content was also observed in the L3-protein or L5-protein group when compared to that in the vehicle group (*p* < 0.0001 and <0.01, Figure 2C,D). There was no significant difference between the L3-protein and L5-protein groups in terms of liver lipid content.

Plaque areas of the aortic arch were also detected by Oil Red O staining of pathological sections. As shown in Figure 1E,F, evident plaques were observed in the model group, indicating successful AS modeling in the present study. There was a significant decrease in relative plaque areas in the *N. brasiliensis* infection group when compared to that in the model group (*p* < 0.01, Figure 1E,F). Meanwhile, a smaller relative plaque area was also observed in the L3-protein or L5-protein groups (*p* < 0.0001, Figure 2E,F) when compared to that in the vehicle group. Relative plaque areas in the L5-protein group were smaller when compared to that in the L3-protein group (*p* < 0.01, Figure 2F).

These results indicated reduced liver lipid content and AS plaque areas during the initiation of AS by *N. brasiliensis* infection and derived products intervention in the ApoE^-/-^ mouse model when fed a high-fat diet.

#### 3.1.3. Serum Cytokine Levels

Serum samples were collected at the end of week 12, and the cytokine levels of each mouse were detected by flow cytometry. As shown in Figure 1G–I, the cytokine levels of IL-6, IL-4, and IL-10 were significantly elevated in the *N. brasiliensis* infection group when compared to those in the model group (*p* < 0.05, 0.01, and 0.05, respectively). The levels of IL-2, IFN-γ, and TNF-α were also detected but showed no significant differences between the two groups (data not shown). When compared to the vehicle group, TNF-α levels in the L3-protein and L5-protein groups were significantly decreased (Figure 2G, *p* < 0.05 and 0.01). IL-6 levels were only significantly increased in the L5-protein group when compared to those in the vehicle group (Figure 2H, *p* < 0.05). Furthermore, there was a significant increase in the levels of IL-4 and IL-10 in the L3-protein and L5-protein groups when compared to those in the vehicle group (*p* < 0.01, Figure 2I,J). Furthermore, there were no significant differences in the levels of TNF-α, IL-6, IL-4, and IL-10 between the L3-protein and L5-protein groups. IL-2 and IFN-γ were also detected in the three groups; no significant differences were observed (data not shown). These results indicated that anti-inflammation effects could be induced through *N. brasiliensis* infection and derived products (L3-protein or L5-protein) intervention during AS initiation in the ApoE^-/-^ mouse model when fed with a high-fat diet.

### 3.2. Effects of N. brasiliensis and Derived Products Intervention on the Progression of AS

#### 3.2.1. Histopathological Alterations in the Liver and Aortic Arch

Oil Red O staining of pathological sections was used to evaluate lipid contents in the liver and plaque areas of the aortic arch. As shown in Figure 3B–E, there were significant reductions in liver lipid content and relative plaque areas in the *N. brasiliensis* infection group when compared to that in the model group (*p* < 0.01). For *N. brasiliensis*-derived products intervention, there were also significant reductions in liver lipid content and relative plaque areas in the L3-protein and L5-protein groups when compared to that in the vehicle group (Figure 4B–E, *p* < 0.01). However, there were no significant differences between the L3-protein and L5-protein groups with regard to liver lipid content and relative plaque areas. These results indicated a protective effect manifesting as decreased liver lipid content and contractible plaque area, was induced by *N. brasiliensis* infection and derived products intervention during the process of AS progression in the ApoE^-/-^ mouse model when fed a high-fat diet.

#### 3.2.2. Serum Lipid Levels

Serum samples were collected from each mouse at the end of week 20, and fasting serum lipid levels were analyzed. As shown in Figure 3F–H, serum lipid levels, including total cholesterol, triglyceride, and low-density lipoprotein, decreased significantly following *N. brasiliensis* infection intervention when compared to the model group (*p* < 0.01, <0.01, and <0.001, respectively). However, a significant elevation of high-density lipoprotein levels was observed following *N. brasiliensis* infection intervention when compared to the model group (Figure 3I, *p* < 0.001). For *N. brasiliensis*-derived products interventions, there were no significant differences in the serum levels of total cholesterol and triglyceride in the L3-protein and L5-protein groups when compared to the vehicle group (Figure 4F,G). However, there was a significant decrease in serum low-density lipoprotein levels in L3-protein and L5-protein groups when compared to those in the vehicle group (Figure 4H, *p* < 0.0001 and <0.001). No significant difference was observed between the L3-protein and L5-protein groups. Furthermore, there was a significant elevation of the serum levels of high-density lipoprotein in the L3-protein group (*p* < 0.05); however, there was no significant difference when compared between the L5-protein group and the vehicle group (Figure 4I). Results indicated that *N. brasiliensis* infection and derived products intervention could reduce the serum levels of lipid (especially low-density protein levels) and increase protective high-density lipoprotein levels during the process of AS progression in the ApoE^-/-^ mouse model fed a high-fat diet.

#### 3.2.3. Serum Cytokine Levels

Serum samples were collected from each mouse at the end of week 20, and serum levels of IL-2, IFN-γ, TNF-α, IL-6, IL-4, and IL-10 were detected by flow cytometry as described above. For *N. brasiliensis* infection intervention, there were significant decreases in the levels of IFN-γ and IL-2 in the *N. brasiliensis* infection group when compared to those in the model group (Figure 3J,K, *p* < 0.01). There were significant elevations of TNF-α, IL-6, and IL-10 levels in the *N. brasiliensis* infection group when compared to those in the model group (Figure 3L–N, *p* < 0.01, <0.05, and <0.0001, respectively). No significant difference in IL-4 levels was observed between the two groups (data not shown). For *N. brasiliensis*-derived products intervention, there were significant decreases of IL-2 and IL-6 levels in the L3-protein and L5-protein groups when compared to the vehicle group (Figure 4J,L, *p* < 0.001, <0.05, <0.05 and <0.001, respectively). Furthermore, there was a significant elevation of IL-10 levels in the L3-protein and L5-protein groups when compared to the vehicle group (Figure 4M, *p* < 0.001 and <0.001). No significant differences between IFN-γ (data not shown) and TNF-α (Figure 4K) were observed between the three groups. These results indicated that *N. brasiliensis* infection and derived products interventions could decrease pro-inflammation cytokine levels (IL-2 and IFN-γ) and increase anti-inflammation cytokine levels (IL-10) during the process of AS progression in the ApoE^-/-^ mouse model fed a high-fat diet.

## 4. Discussion

It has been widely accepted that helminth infection or helminth-derived products could be beneficial for inflammatory diseases and might provide a novel approach to the prevention or treatment of disease [11,27,28,29]. As one type of low-grade chronic inflammatory disease, AS is known to be accompanied by proinflammatory responses during different stages of AS development [1,30]. Therapeutic approaches targeting the inhibition of inflammation have proved to be effective for AS prevention and alleviation in previous studies [2,7,31]. Hookworm infection could induce a wide range of type two protective immune responses for resident tissue repair and anti-inflammatory responses, which could provide a novel remedy for inflammatory disease intervention, such as for AS [23,24]. Based on these concepts, we designed and carried out the present study through *N. brasiliensis* infection and derived product intervention (immunization) for AS in the ApoE^-/-^ mouse model for the first time. Results showed that *N. brasiliensis* infection and derived products intervention could prevent and alleviate the development of AS, manifesting as lower weight increases, smaller areas of AS plaques, lower levels of lipid content in the liver and aortic arch and higher levels of anti-inflammation cytokines in the serum.

Direct helminth infection, also known as worm therapy, was first proposed in 1968, thus demonstrating their importance for human health [32]. Numerous experiments indicated the safety of worm therapy following infection with different types of helminths, including pig whipworm (*Trichuris suis*), human hookworm (*Necator americanus*), and rat tapeworm (*Hymenolepis diminuta*), thus demonstrating a therapeutic role in autoimmune, inflammatory, and allergic diseases [33,34,35]. In the present study, we used *N. brasiliensis* infection as a model of human hookworm to intervene in AS and observed a protective role for AS initiation and progression. Due to self-cure from *N. brasiliensis* infection within two weeks in the mouse model, we carried out multiple infections to sustain the infection conditions in our two animal experiments. Nonetheless, we still observed a weakened physical condition in the infected mice (data not shown), which may have influenced the final identification of protective effects. For safety and ethical consideration, screening the helminth-derived molecules with biological activities would be more suitable for worm therapy in the future; a variety of helminth-derived molecules have been screened and shown to exert different biological activities for disease intervention [27,36]. In the present study, we observed intervention effects for AS initiation and progression by using *N. brasiliensis*-derived products, L3- and L5-derived proteins, through a typical immunization procedure; these had a similar protective effect to *N. brasiliensis* infection, thus providing possibilities for screening and evaluation helminth-derived biologically active molecules for AS intervention in the future. However, translation from rodent hookworm to human hookworm is another key consideration in subsequent research. Furthermore, *N. brasiliensis*-derived excretory-secretory products could be another important source for active molecule screening; these contain various anti-inflammatory metabolites, as demonstrated previously; these could be investigated for AS intervention in the near future [26,37].

Mechanisms of AS initiation and progression have not been fully elucidated until now, although low-grade chronic inflammation was thought to play an important role [38]. Anti-inflammation therapeutic approaches, including anti-inflammatory drugs, biological therapies targeting cytokines and chemokines, and small molecule enzyme inhibitors and receptor antagonists, have all been attempted and have been shown to exert efficient antiatherosclerosis roles [6,39,40]. According to the present study, increased anti-inflammatory cytokine levels, including IL-4 and IL-10, were observed. Furthermore, we also observed decreased levels of proinflammatory cytokine levels, including IL-2, TNF-α, and IFN-γ, in serum samples, thus contributing to the final contractible plaque areas. However, the high levels of IL-10 and IL-4 and reduced levels of proinflammatory cytokines observed in the present study probably also reflect immunoregulation, which needs to be further verified. Furthermore, an in-depth study should be carried out to investigate the local distribution of inflammatory cytokines and major cytokine-secreting cell types within the plaques. As reported previously, *N. brasiliensis*-derived products (L3 and L5 proteins) could induce M2 macrophage polarization in vitro [25]; whether M2 macrophage polarization status could also be induced by L3 and L5 proteins and play an anti-inflammatory activity in vivo needs to be investigated further. Interleukin-6 (IL-6), a unique pleiotropic cytokine, exhibits both pro- and anti-inflammatory properties depending on different cell types and animal models [41,42]; this could explain the alterations of serum IL-6 levels observed in the present study.

High serum lipoprotein levels, especially elevated low-density lipoprotein (LDL) levels, are known to be strongly associated with AS development [43,44]. AS lesions were found to be initiated via the enhanced uptake of LDL by monocytes and macrophages [45,46]. For classical therapies, the lipid-lowering therapeutic approach is still applied for AS interventions, including statins and fibrates [47,48]. In the present study, we also observed a reduction in lipid content in the liver and aortic arch following *N. brasiliensis* infection and derived products intervention. Furthermore, decreased levels of serum lipoprotein were observed, including total cholesterol, triglyceride, and low-density lipoprotein, following intervention; this indicated that *N. brasiliensis* and derived products could reduce serum lipoprotein. In contrast to LDL, high-density lipoprotein (HDL) is considered to be beneficial for blood cholesterol and can protect against AS [49]. Our results also showed an elevation in serum HDL levels following *N. brasiliensis* and derived products intervention; this could further induce the protective effects against AS development. However, the mechanisms associated with the alterations of serum lipoprotein levels and different interventions need to be further investigated.

In summary, a protective effect was confirmed for atherosclerosis initiation and progression by *N. brasiliensis* and derived product intervention in a mouse model, possibly via anti-inflammatory and lipid-lowering approaches, thus providing new insight for the prevention and treatment of atherosclerosis.

## Figures and Tables

**Figure 1 pathogens-11-01208-f001:**
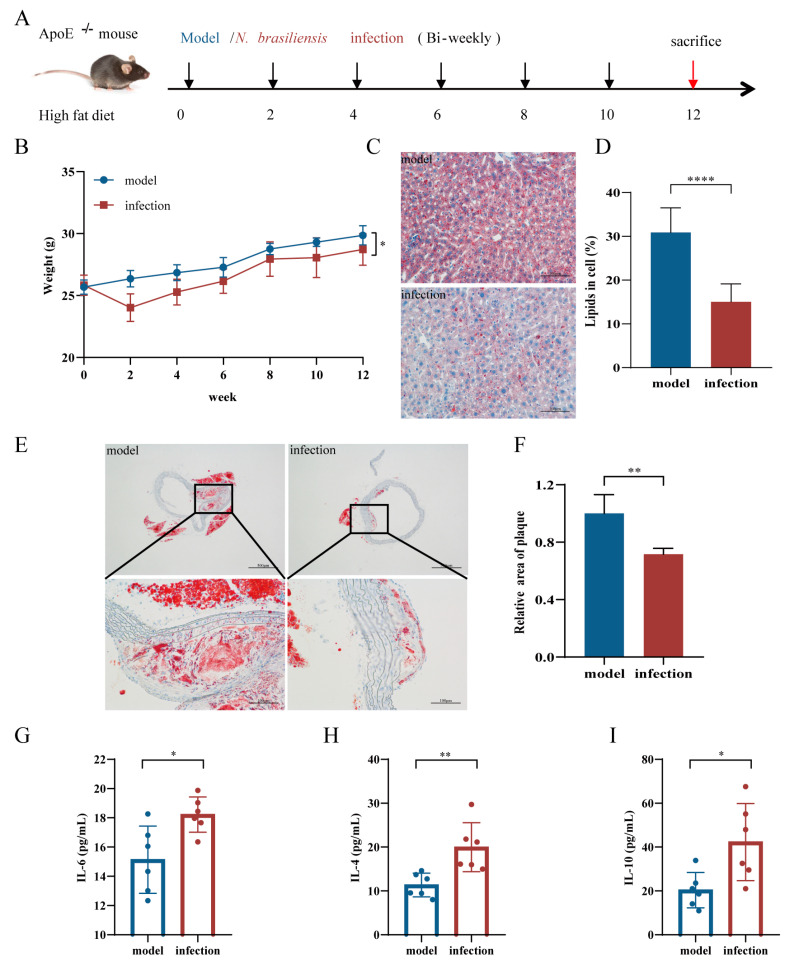
*N. brasiliensis* infection could prevent atherosclerosis. (**A**): Schematic diagram of the experimental design through *N. brasiliensis* infection intervention in atherosclerosis. (**B**): Body weight changes in mice in the model (n = 6) and *N. brasiliensis* infection groups (n = 6) during the initial stages of atherosclerosis; (**C**,**D**): Representative images of Oil Red O Staining of liver and quantitative data relating to lipid percentage in mice from the model group (n = 6) and *N. brasiliensis* infection group (n = 6); (**E**,**F**): Representative images of Oil Red O Staining of the aortic roots of mice in the model (n = 6) and *N. brasiliensis*-infected groups (n = 6) and quantitative data relating to relative plaque area. (**G**–**I**): Serum cytokine levels in the model group (n = 6) and *N. brasiliensis*-infected group (n = 6). The “n” indicates the number of biological replicates. *p* < 0.05 considered significant; ns = not significant, ^*^
*p* < 0.05; ^**^
*p* < 0.01; ^****^
*p* < 0.0001 as determined by Student’s *t*-test comparisons of two groups or one-way ANOVA followed by Dunnett multiple comparisons test to compare more than two groups. Error bars represent SEMs.

**Figure 2 pathogens-11-01208-f002:**
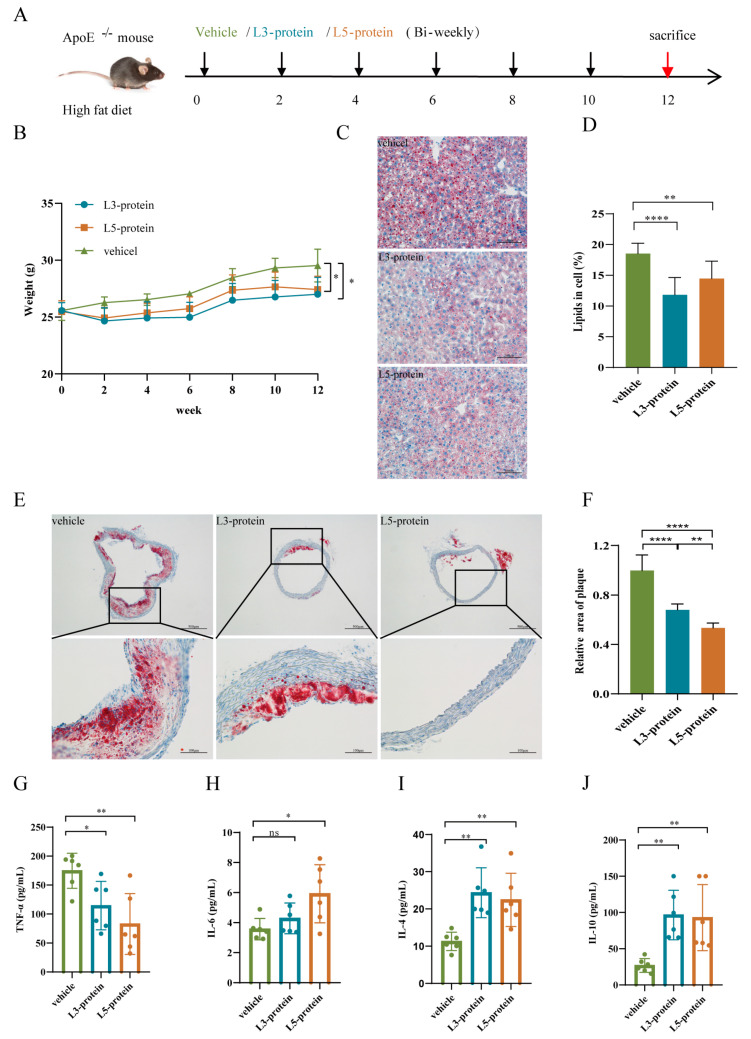
*N. brasiliensis*-derived proteins could prevent atherosclerosis. (**A**): Schematic diagram of the experimental design of *N. brasiliensis*-derived protein intervention in atherogenesis. (**B**): Body weight changes in vehicle (n = 6)-, L3-protein (n = 6)-, and L5-protein (n = 6)-treated mice during the initial stages of atherosclerosis; (**C**,**D**): Representative images of Oil Red O Staining and quantitative data of lipid percentage in the liver of vehicle (n = 6)-, L3-protein (n = 6)-, and L5-protein (n = 6)-treated mice; (**E**,**F**): Representative images of Oil Red O Staining and quantitative data on relative plaque area in aortic roots of vehicle (n = 6)-, L3-protein (n = 6)-, and L5-protein (n = 6)-treated mice; (**G**–**J**): Serum cytokine levels in mice treated with vehicle (n = 6), L3-protein (n = 6), and L5-protein (n = 6). The “n” indicates the number of biological replicates. *p* < 0.05 considered significant; ns = not significant, ^*^
*p* < 0.05; ^**^
*p* < 0.01; ^****^
*p* < 0.0001 as determined by Student’s *t*-test comparisons of two groups or one-way ANOVA followed by Dunnett multiple comparisons test to compare more than two groups. Error bars represent SEMs.

**Figure 3 pathogens-11-01208-f003:**
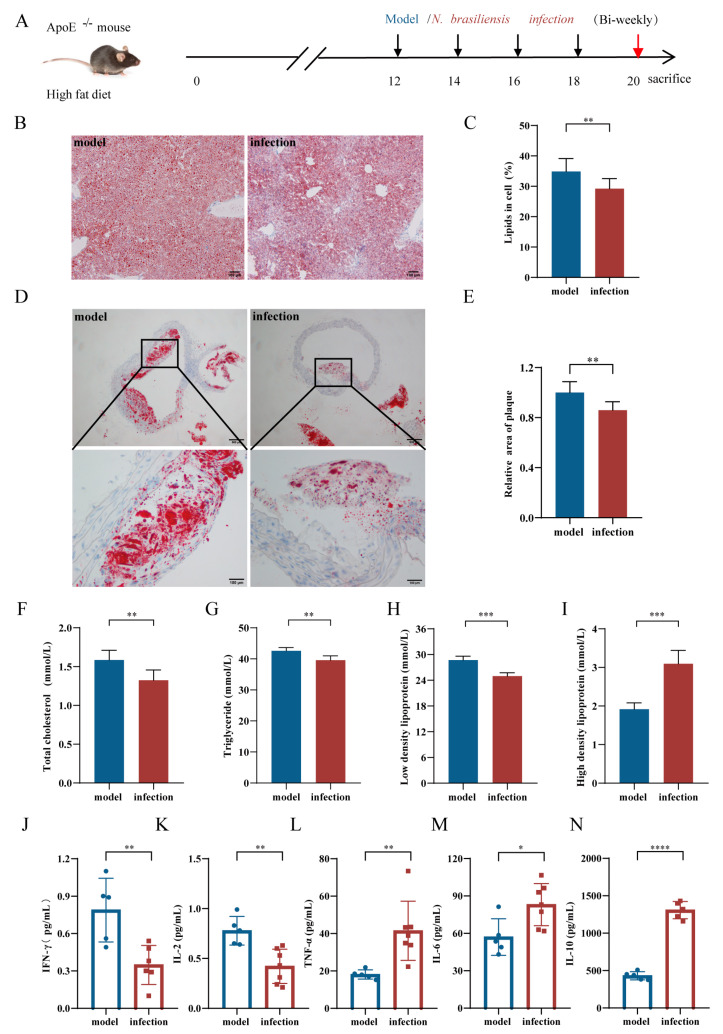
*N. brasiliensis* infection slowed the progression of atherosclerosis. (**A**): Schematic diagram of the experimental design of *N. brasiliensis* infection to intervene in the progression of atherosclerosis. (**B**,**C**): Representative images of Oil Red O Staining of liver and quantitative data of lipid percentage in model (n = 5) and *N. brasiliensis*-infected (n = 5–7) mice; (**D**,**E**): Representative images of Oil Red O Staining of aortic roots and quantitative data of relative plaque area in model (n = 5) and *N. brasiliensis*-infected (n = 5–7) mice. (**F**–**I**): Serum levels of total cholesterol, triglycerides, LDL, and HDL in the model group (n = 5) and *N. brasiliensis*-infected group (n = 5–7) mice; (**J**–**N**): Serum levels of cytokines in the model group (n = 5) and *N. brasiliensis*-infected group (n = 6–7). The “n” indicates the number of biological replicates. *p* < 0.05 considered significant; ns = not significant, ^*^
*p* < 0.05; ^**^
*p* < 0.01; ^***^
*p* < 0.001; ^****^
*p* < 0.0001 as determined by Student’s *t*-test comparisons of two groups or one-way ANOVA followed by Dunnett multiple comparisons test to compare more than two groups. Error bars represent SEMs.

**Figure 4 pathogens-11-01208-f004:**
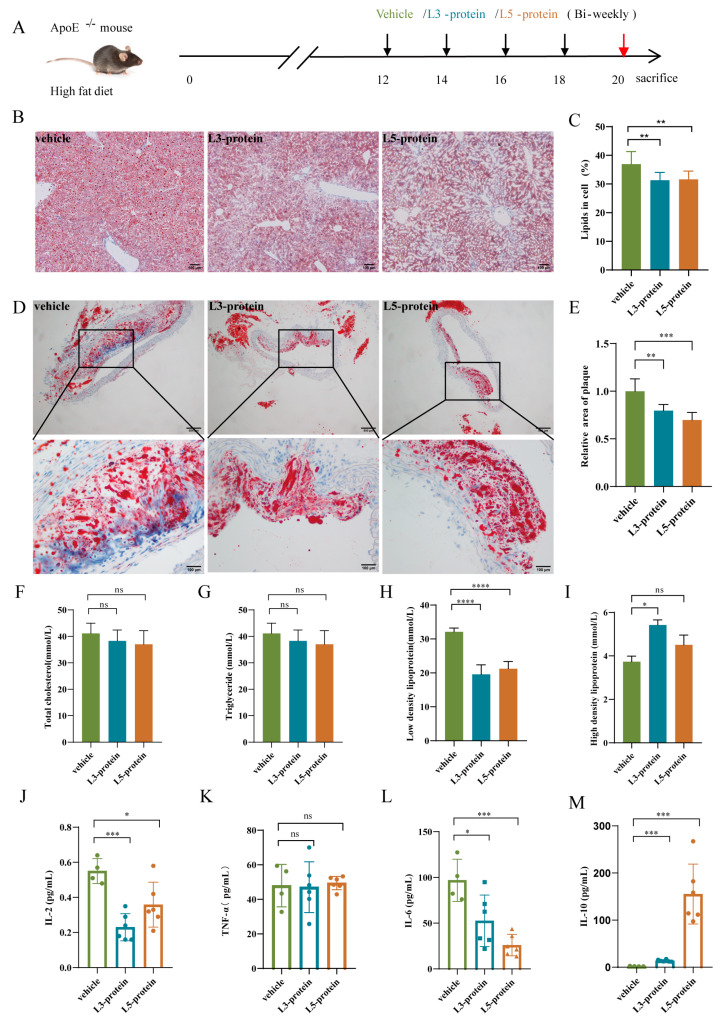
*N. brasiliensis*-derived proteins slowed the progression of atherosclerosis. (**A**): Schematic diagram of the experimental design of *N. brasiliensis*-derived protein intervention in atherosclerosis progression. (**B**,**C**): Representative images of Oil Red O Staining of liver and quantitative data of lipid percentage in vehicle (n = 4)-, L3-protein (n = 6)-, and L5-protein (n = 6)-treated mice; (**D**,**E**): Representative images of Oil Red O Staining of aortic roots and quantitative data of relative plaque area in vehicle (n = 6)-, L3-protein (n = 6)-, and L5-protein (n = 6)-treated mice; (**F**–**I**): Total cholesterol, triglyceride, LDL and HDL levels in serum of vehicle(n = 6)-, L3-protein(n = 6)-, and L5-protein(n = 6)-treated mice. (**J**–**M**): Cytokine levels in the serum of vehicle (n = 6)-, L3 protein (n = 6)-, and L5 protein (n = 6)-treated mice. The “n” indicates the number of biological replicates. *p* < 0.05 considered significant; ns = not significant, ^*^
*p* < 0.05; ^**^
*p* < 0.01; ^***^
*p* < 0.001; ^****^
*p* < 0.000 1 as determined by Student’s *t*-test comparisons of two groups or one-way ANOVA followed by Dunnett multiple comparisons test to compare more than two groups. Error bars represent SEMs.

## Data Availability

Not applicable.

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
