# Peer review of "Inhibition Effects of Nippostrongylus brasiliensis and Its Derivatives against Atherosclerosis in ApoE-/- Mice through Anti-Inflammatory Response"

_pathogens, 2022, doi:10.3390/pathogens11101208_

Round 1

Reviewer 1 Report

This is an interesting study that can provide a new insight for the management of Atherosclerosis. The use of N. braziliensis and its derived products is a good idea. I liked the choice of a parasite that finishes its cycle in the host spontaneously. Results are appropriate and are very promising, and most of the references are very recent.  English language and style are check required

Author Response

Thanks for the reviewer’s comments. And we have already checked the language and style in manuscript with revisions mode.

Reviewer 2 Report

This was a good piece of research which was well presented.  As suggested above, there should be minor edits to the English

Author Response

(The authors gave the same response as above.)

Reviewer 3 Report

The anti-inflammatory activity of full nematode infection and its larval stage proteins were measured. N. brasiliensis infection and immunization with two stages (L3 and L5) larvae proteins were used to reduce atherosclerosis in a mouse model. The authors decided to monitor histopathology in aortic arch plaque areas, liver lipid contents, and cytokine response in ApoE-/- mice maintained on a high-fat diet.

Major remarks

 The authors claim that they evaluated the anti-inflammatory activity of nematode infection and the nematode products. However, the methodology responded only to the statement that several infections of mice in two weeks of intervals resulted in reduced pathology in atherosclerosis in mice and that immunisation of mice with the nematode proteins might reduce pathology in a proposed model of atherosclerosis. The author induced immune response in mice by infection with N. brasiliensis and by the nematode proteins.

 The obtained results present an immune environment created by immunisation but not the biological activity of nematode proteins, at least after immunisation with N. brasiliensis proteins. Using proteins with Freund adjuvant is a typical procedure for induction of immunity to selected antigens, but not to keep these biological activities, in that case, anti-inflammatory. It would be worth to mention about it in the discussion. Do you check the number of nematodes present in the intestine of several infected mice? Primary induced immune response might be active against the larvae stage in the lung. The aim of the studies does not clearly explain the possible influence of mice's pattern of immune response to N. brasiliensis infection in atherosclerotic circumstances.

 Single infection with N. brasiliensis induces high Th2-related hypersensitive inflammation resulting in a self-cure. Multiple infections for a prolonged time are not a case of maintaining primary infection conditions; these were immunization with the alive L3 stage.

Several infections of mice with alive larvae or inoculation with nematode proteins might induce a Th2/Treg immune response to the nematode antigens. The high levels of IL-10 and IL-4 and reduced levels of proinflammatory cytokines observed in the studies probably reflect immunoregulation.

Minor remarks 

 In materials and methods, eight mice consisted in each experimental group, but a different number of animals: e.g., 5-7 represent each group in the results. What was the reason for that difference?

Please indicate how many times mice were infected with L3 and how many times mice were injected with proteins and oil emulsion. Do the arrows on the graphs indicate the point (week) of inoculation? If it is ok, why were mice infected on the day of final sampling? (Fig 3., Fig 4). The pattern of measured cytokines in each experiment is also distinct, and no possibility to compare the similar pattern of the immune response between them, even when the statistical difference was not significant, exists.

 In N. brasiliens life cycle, L5 is a pre-adult stage before the copulation of nematodes. Therefore L5 stage larvae are not the same as the adult form in the patent phase of infection. The difference is in the content of egg antigen, typical for fertile females who lay eggs. 

Other notes are available in the manuscript and are highlighted.

Author Response

We have responded in the following document

Reviewer 4 Report

Yougui Yang and colleagues in their manuscript entitled ‘Inhibition effects of Nippostrongylus brasiliensis and its derivatives against atherosclerosis in ApoE-/- mice through anti-inflammatory response’ investigated the effective regulatory role of N. brasiliensis and its derived proteins in the development and progression of atherosclerosis. Although this study is interesting there are several caveats to trusting the data whether it is leading to the protective effects. It is a well-written manuscript, however, with regards to the data, there are few more confirmations required. I have provided my major and minor comments below.

1. Maintainance of N. brasiliensis needs to be added in the methods section. There is no clear indication of the effectiveness of each dosage before it was analyzed for the atherosclerosis model study.

2. Please elaborate in the methods section on how the lipids in cells were quantified. A Sirius red stain should also be performed if there is neointimal fibrosis.

3. It is hard to understand if the ApoE is the relevant model in this infection study since the authors have not included any WT mice as an additional control group.

4. To verify that these mice actually lack ApoE expression after infection, authors must include ApoE expression.

5. In all the histology sections and the staining for Oil-red O, why most of the mice models with the infection showed plaques only in some areas compared to their uninfected controls? Please provide representative images.

6. To confirm that serum levels are corroborated at the vessel wall pathology, immunostainings of the relevant cytokines are extremely important.

7. Minor comment: spell checking of the figures is required, for example, fig2B. Also, why are the thickness of each vessel wall looks different in the treated models?

8. Each model pursued in this study should provide blood lipid data in order to understand the empirical effects observed. 

Author Response

We have responded in the following document.

Round 2

Reviewer 3 Report

I find the munuscript improved, however, still needs better clarification of the hypothesis. Therefore  I suggest correcting the sentence in the abstract at line 23 and in the introduction at lines 98- 99. Please replace the sentence with: ".... intervention with N. brasiliensis infection and immunization with its derived products, we analyzed several factors to evaluate interventional effects.....". 

Author Response

I find the manuscript improved, however, still needs better clarification of the hypothesis. Therefore, I suggest correcting the sentence in the abstract at line 23 and in the introduction at lines 98- 99. Please replace the sentence with: ".... intervention with N. brasiliensis infection and immunization with its derived products, we analyzed several factors to evaluate interventional effects.....".

Response: Thanks for the reviewer’s suggestions. We have revised in the manuscript according to the reviewer’s suggestions (Page 1, Lines 21–23, and Page 3, Lines101–105).

Reviewer 4 Report

I think the authors have improved the manuscript and addressed all the queries posed by this reviewer. 

Author Response

I think the authors have improved the manuscript and addressed all the queries posed by this reviewer.

Response: Thanks again to the reviewer for the kindly suggestions.